# Estimating Mixture Models via Mixtures of Polynomials

**Sida I. Wang**    **Arun Tejasvi Chaganty**    **Percy Liang**
Computer Science Department, Stanford University, Stanford, CA, 94305
`{sidaw,chaganty,pliang}@cs.stanford.edu`

## Abstract

Mixture modeling is a general technique for making any simple model more expressive through weighted combination. This generality and simplicity in part explains the success of the Expectation Maximization (EM) algorithm, in which updates are easy to derive for a wide class of mixture models. However, the likelihood of a mixture model is non-convex, so EM has no known global convergence guarantees. Recently, method of moments approaches offer global guarantees for some mixture models, but they do not extend easily to the range of mixture models that exist. In this work, we present Polymom, an unifying framework based on method of moments in which estimation procedures are easily derivable, just as in EM. Polymom is applicable when the moments of a single mixture component are polynomials of the parameters. Our key observation is that the moments of the mixture model are a mixture of these polynomials, which allows us to cast estimation as a Generalized Moment Problem. We solve its relaxations using semidefinite optimization, and then extract parameters using ideas from computer algebra. This framework allows us to draw insights and apply tools from convex optimization, computer algebra and the theory of moments to study problems in statistical estimation. Simulations show good empirical performance on several models.

## 1 Introduction

Mixture models play a central role in machine learning and statistics, with diverse applications including bioinformatics, speech, natural language, and computer vision. The idea of mixture modeling is to explain data through a weighted combination of simple parametrized distributions [1, 2]. In practice, maximum likelihood estimation via Expectation Maximization (EM) has been the workhorse for these models, as the parameter updates are often easily derivable. However, EM is well-known to suffer from local optima. The method of moments, dating back to Pearson [3] in 1894, is enjoying a recent revival [4, 5, 6, 7, 8, 9, 10, 11, 12, 13] due to its strong global theoretical guarantees. However, current methods depend strongly on the specific distributions and are not easily extensible to new ones.

In this paper, we present a method of moments approach, which we call Polymom, for estimating a wider class of mixture models in which the moment equations are polynomial equations (Section 2). Solving general polynomial equations is NP-hard, but our key insight is that for mixture models, the moments equations are *mixtures of polynomials equations* and we can hope to solve them if the moment equations for each mixture component are *simple* polynomials equations that we can solve. Polymom proceeds as follows: First, we recover mixtures of monomials of the parameters from the data moments by solving an instance of the Generalized Moment Problem (GMP) [14, 15] (Section 3). We show that for many mixture models, the GMP can be solved with basic linear algebra and in the general case, can be approximated by an SDP in which the moment equations are linear constraints. Second, we extend multiplication matrix ideas from the computer algebra literature [16,

| mixture model | | sizes | |
|---|---|---|---|
| $\mathbf{x}_t$ | data point ($\mathbb{R}^D$) | $D$ | data dimensions |
| $z_t$ | latent mixture component ($[K]$) | $K$ | mixture components |
| $\boldsymbol{\theta}_k$ | parameters of component $k$ ($\mathbb{R}^P$) | $P$ | parameters of mixture components |
| $\pi_k$ | mixing proportion of $p(z=k)$ | $T$ | data points |
| $[\boldsymbol{\theta}_k]_{k=1}^K$ | all model parameters | $N$ | constraints |
| **moments of data** | | $[N]$ | $\{1,\dots,N\}$ |
| $\phi_n(\mathbf{x})$ | observation function | $r$ | degree of the moment matrix |
| $f_n(\boldsymbol{\theta})$ | observation function | $s(r)$ | size of the degree $r$ moment matrix |
| **moments of parameters** | | **polynomials** | |
| $\mathscr{L}_\mathbf{y}$ | the Riesz linear functional | $\mathbb{R}[\boldsymbol{\theta}]$ | polynomial ring in variables $\boldsymbol{\theta}$ |
| $y_{\boldsymbol{\alpha}}$ | $y_{\boldsymbol{\alpha}} = \mathscr{L}_\mathbf{y}(\boldsymbol{\theta}^{\boldsymbol{\alpha}})$, $\boldsymbol{\alpha}^{\text{th}}$ moment | $\mathbb{N}$ | set of non-negative integers |
| $\mu$ | probability measure for $\mathbf{y}$ | $\boldsymbol{\alpha},\boldsymbol{\beta},\boldsymbol{\gamma}$ | vector of exponents (in $\mathbb{N}^P$ or $\mathbb{N}^D$) |
| $\mathbf{y}$ | $(y_{\boldsymbol{\alpha}})_{\boldsymbol{\alpha}}$ the moment sequence | $\boldsymbol{\theta}^{\boldsymbol{\alpha}}$ | monomial $\prod_{p=1}^P \theta_p^{\boldsymbol{\alpha}_p}$ |
| $\mathbf{M}_r(\mathbf{y})$ | moment matrix of degree $r$ | $a_{n\boldsymbol{\alpha}}$ | coefficient of $\boldsymbol{\theta}^{\boldsymbol{\alpha}}$ in $f_n(\boldsymbol{\theta})$ |

Table 1: **Notation**: We use lowercase letters (e.g., $d$) for indexing, and the corresponding uppercase letter to denote the upper limit (e.g., $D$, in "sizes"). We use lowercase letters (e.g., $\theta_{k,p}$) for scalars, lowercase bold letters (e.g., $\boldsymbol{\theta}$) for vectors, and bold capital letters (e.g., $\mathbf{M}$) for matrices.

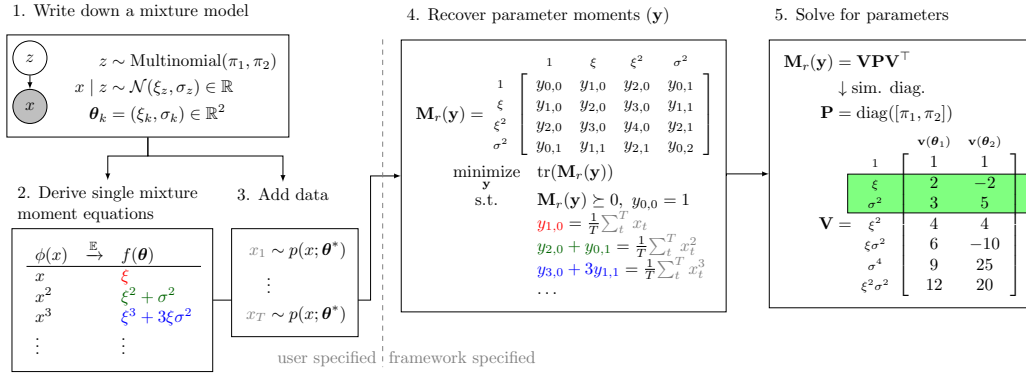

Figure 1: An overview of applying the Polymom framework.

17, 18, 19] to extract the parameters by solving a certain generalized eigenvalue problem (Section 4).

Polymom improves on previous method of moments approaches in both generality and flexibility. First, while tensor factorization has been the main driver for many of the method of moments approaches for many types of mixture models, [6, 20, 9, 8, 21, 12], each model required specific adaptations which are non-trivial even for experts. In contrast, Polymom provides a unified principle for tackling new models that is as turnkey as computing gradients or EM updates. To use Polymom (Figure 1), one only needs to provide a list of observation functions ($\phi_n$) and derive their expected values expressed symbolically as polynomials in the parameters of the specified model ($f_n$). Polymom then estimates expectations of $\phi_n$ and outputs parameter estimates of the specified model. Since Polymom works in an optimization framework, we can easily incorporate constraints such as non-negativity and parameter tying which is difficult to do in the tensor factorization paradigm. In simulations, we compared Polymom with EM and tensor factorization and found that Polymom performs similarly or better (Section 5). This paper assumes identifiability and infinite data. With the exception of a few specific models in Section 5, we defer issues of general identifiability and sample complexity to future work.

## 2  Problem formulation

### 2.1  The method of moments estimator

In a mixture model, each data point $\mathbf{x} \in \mathbb{R}^D$ is associated with a latent component $z \in [K]$:

$$z \sim \text{Multinomial}(\pi), \quad \mathbf{x} \mid z \sim p(\mathbf{x}; \boldsymbol{\theta}_z^*), \tag{1}$$

where $\pi = (\pi_1, \ldots, \pi_K)$ are the mixing coefficients, $\boldsymbol{\theta}_k^* \in \mathbb{R}^P$ are the true model parameters for the $k^{\text{th}}$ *mixture component*, and $\mathbf{x} \in \mathbb{R}^D$ is the random variable representing data. We restrict our attention to mixtures where each component distribution comes from the same parameterized family. For example, for a mixture of Gaussians, $\boldsymbol{\theta}_k^* = (\xi_k^* \in \mathbb{R}^D, \Sigma_k^* \in \mathbb{R}^{D \times D})$ consists of the mean and covariance of component $k$.

We define $N$ *observation functions* $\phi_n : \mathbb{R}^D \to \mathbb{R}$ for $n \in [N]$ and define $f_n(\boldsymbol{\theta})$ to be the expectation of $\phi_n$ over a single component with parameters $\boldsymbol{\theta}$, which we assume is a simple polynomial:

$$f_n(\boldsymbol{\theta}) := \mathbb{E}_{\mathbf{x} \sim p(\mathbf{x}; \boldsymbol{\theta})}[\phi_n(\mathbf{x})] = \sum_{\boldsymbol{\alpha}} a_{n\boldsymbol{\alpha}} \boldsymbol{\theta}^{\boldsymbol{\alpha}}, \tag{2}$$

where $\boldsymbol{\theta}^{\boldsymbol{\alpha}} = \prod_{p=1}^{P} \theta_p^{\alpha_p}$. The expectation of each observation function $\mathbb{E}[\phi_n(\mathbf{x})]$ can then be expressed as a mixture of polynomials of the true parameters $\mathbb{E}[\phi_n(\mathbf{x})] = \sum_{k=1}^{K} \pi_k \mathbb{E}[\phi_n(\mathbf{x})|z = k] = \sum_{k=1}^{K} \pi_k f_n(\boldsymbol{\theta}_k^*)$.

The method of moments for mixture models seeks parameters $[\boldsymbol{\theta}_k]_{k=1}^K$ that satisfy the *moment conditions*

$$\mathbb{E}[\phi_n(\mathbf{x})] = \sum_{k=1}^{K} \pi_k f_n(\boldsymbol{\theta}_k). \tag{3}$$

where $\mathbb{E}[\phi_n(\mathbf{x})]$ can be estimated from the data: $\frac{1}{T}\sum_{t=1}^{T} \phi_n(\mathbf{x}_t) \xrightarrow{p} \mathbb{E}[\phi_n(\mathbf{x})]$. The goal of this work is to find parameters satisfying moment conditions that can be written in the mixture of polynomial form (3). We assume that the $N$ observations functions $\phi_1, \ldots, \phi_N$ uniquely identify the model parameters (up to permutation of the components).

**Example 2.1** (1-dimensional Gaussian mixture). *Consider a $K$-mixture of 1D Gaussians with parameters $\boldsymbol{\theta}_k = [\xi_k, \sigma_k^2]$ corresponding to the mean and variance, respectively, of the $k$-th component (Figure 1: steps 1 and 2). We choose the observation functions, $\phi(x) = [x^1, \ldots, x^6]$, which have corresponding moment polynomials, $f(\boldsymbol{\theta}) = [\xi, \xi^2 + \sigma^2, \xi^3 + 3\xi\sigma^2, \ldots]$. For example, instantiating (3), $\mathbb{E}[x^2] = \sum_{k=1}^{K} \pi_k(\xi_k^2 + \sigma_k^2)$. Given $\phi(\mathbf{x})$ and $f(\boldsymbol{\theta}^*)$, and data, the Polymom framework can recover the parameters. Note that the 6 moments we use have been shown by [3] to be sufficient for a mixture of two Gaussians.*

**Example 2.2** (Mixture of linear regressions). *Consider a mixture of linear regressions [22, 9], where each data point $\mathbf{x} = [x, y]$ is drawn from component $k$ by sampling $x$ from an unknown distribution independent of $k$ and setting $y = w_k x + \epsilon$, where $\epsilon \sim \mathcal{N}(0, \sigma_k^2)$. The parameters $\boldsymbol{\theta}_k = (w_k, \sigma_k^2)$ are the slope and noise variance for each component $k$. Let us take our observation functions to be $\phi(\mathbf{x}) = [x, xy, xy^2, x^2, \ldots, x^3y^2]$, for which the moment polynomials are $f(\boldsymbol{\theta}) = [\mathbb{E}[x], \mathbb{E}[x^2]w, \mathbb{E}[x^3]w^2 + \mathbb{E}[x]\sigma^2, \mathbb{E}[x^2], \ldots]$.*

In Example 2.1, the coefficients $a_{n\boldsymbol{\alpha}}$ in the polynomial $f_n(\boldsymbol{\theta})$ are just constants determined by integration. For the conditional model in Example 2.2, the coefficients depends on the data. However, we cannot handle arbitrary data dependence, see Section D for sufficient conditions and counterexamples.

### 2.2  Solving the moment conditions

Our goal is to recover model parameters $\boldsymbol{\theta}_1^*, \ldots, \boldsymbol{\theta}_K^* \in \mathbb{R}^P$ for each of the $K$ components of the mixture model that generated the data as well as their respective mixing proportions $\pi_1, \ldots, \pi_K \in \mathbb{R}$. To start, let's ignore sampling noise and identifiability issues and suppose that we are given exact moment conditions as defined in (3). Each condition $f_n \in \mathbb{R}[\boldsymbol{\theta}]$ is a polynomial of the parameters $\boldsymbol{\theta}$, for $n = 1, \ldots, N$.

Equation 3 is a polynomial system of $N$ equations in the $K + K \times P$ variables $[\pi_1, \ldots, \pi_K]$ and $[\boldsymbol{\theta}_1, \ldots, \boldsymbol{\theta}_K] \in \mathbb{R}^{P \times K}$. It is natural to ask if standard polynomial solving methods can solve (3) in the case where each $f_n(\boldsymbol{\theta})$ is simple. Unfortunately, the complexity of general polynomial equation solving is lower bounded by the number of solutions, and each of the $K!$ permutations of the mixture components corresponds to a distinct solution of (3) under this polynomial system representation. While several methods can take advantage of symmetries in polynomial systems [23, 24], they still cannot be adapted to tractably solve (3) to the best of our knowledge.

The key idea of Polymom is to exploit the mixture representation of the moment equations (3). Specifically, let $\mu^*$ be a particular "mixture" over the component parameters $\boldsymbol{\theta}_1^*, \ldots, \boldsymbol{\theta}_k^*$ (i.e. $\mu^*$ is a probability measure). Then we can express the moment conditions (3) in terms of $\mu^*$:

$$\mathbb{E}[\phi_n(\mathbf{x})] = \int f_n(\boldsymbol{\theta}) \, \mu^*(d\boldsymbol{\theta}), \text{ where } \mu^*(\boldsymbol{\theta}) = \sum_{k=1}^{K} \pi_k \delta(\boldsymbol{\theta} - \boldsymbol{\theta}_k^*). \tag{4}$$

As a result, solving the original moment conditions (3) is equivalent to solving the following feasibility problem over $\mu$, but where we deliberately "forget" the permutation of the components by using $\mu$ to represent the problem:

$$\begin{aligned} \text{find} \quad & \mu \in \mathcal{M}_+(\mathbb{R}^P), \text{ the set of probability measures over } \mathbb{R}^P \\ \text{s.t.} \quad & \int f_n(\boldsymbol{\theta}) \, \mu(d\boldsymbol{\theta}) = \mathbb{E}[\phi_n(\mathbf{x})], \quad n = 1, \ldots, N \\ & \mu \text{ is } K\text{-atomic (i.e. sum of } K \text{ deltas).} \end{aligned} \tag{5}$$

If the true model parameters $[\boldsymbol{\theta}_k^*]_{k=1}^K$ can be identified by the $N$ observed moments up to permutation, then the measure $\mu^*(\boldsymbol{\theta}) = \sum_{k=1}^{K} \pi_k \delta(\boldsymbol{\theta} - \boldsymbol{\theta}_k^*)$ solving Problem 5 is also unique.

Polymom solves Problem 5 in two steps:

1. Moment completion (Section 3): We show that Problem 5 over the measure $\mu$ can be relaxed to an SDP over a certain (parameter) *moment matrix* $\mathbf{M}_r(\mathbf{y})$ whose optimal solution is $\mathbf{M}_r(\mathbf{y}^*) = \sum_{k=1}^{K} \pi_k \mathbf{v}_r(\boldsymbol{\theta}_k^*) \mathbf{v}_r(\boldsymbol{\theta}_k^*)^\top$, where $\mathbf{v}_r(\boldsymbol{\theta}_k^*)$ is the vector of all monomials of degree at most $r$.

2. Solution extraction (Section 4): We then take $\mathbf{M}_r(\mathbf{y})$ and construct a series of generalized eigendecomposition problems, whose eigenvalues yield $[\boldsymbol{\theta}_k^*]_{k=1}^K$.

**Remark.** From this point on, distributions and moments refer to $\mu^*$ which is over *parameters*, not over the data. All the structure about the data is captured in the moment conditions (3).

## 3 Moment completion

The first step is to reformulate Problem 5 as an instance of the Generalized Moment Problem (GMP) introduced by [15]. A reference on the GMP, algorithms for solving GMPs, and its various extensions is [14]. We start by observing that Problem 5 really only depends on the integrals of monomials under the measure $\mu$: for example, if $f_n(\boldsymbol{\theta}) = 2\theta_1^3 - \theta_1^2 \theta_2$, then we only need to know the integrals over the constituent monomials ($y_{3,0} := \int \theta_1^3 \mu(d\boldsymbol{\theta})$ and $y_{2,1} := \int \theta_1^2 \theta_2 \mu(d\boldsymbol{\theta})$) in order to evaluate the integral over $f_n$. This suggests that we can optimize over the (parameter) *moment sequence* $\mathbf{y} = (y_{\boldsymbol{\alpha}})_{\boldsymbol{\alpha} \in \mathbb{N}^P}$, rather than the measure $\mu$ itself. We say that the moment sequence $\mathbf{y}$ has a *representing measure* $\mu$ if $y_{\boldsymbol{\alpha}} = \int \boldsymbol{\theta}^{\boldsymbol{\alpha}} \, \mu(d\boldsymbol{\theta})$ for all $\boldsymbol{\alpha}$, but we do not assume that such a $\mu$ exists. The *Riesz linear functional* $\mathcal{L}_{\mathbf{y}} : \mathbb{R}[\boldsymbol{\theta}] \to \mathbb{R}$ is defined to be the linear map such that $\mathcal{L}_{\mathbf{y}}(\boldsymbol{\theta}^{\boldsymbol{\alpha}}) := y_{\boldsymbol{\alpha}}$ and $\mathcal{L}_{\mathbf{y}}(1) = 1$. For example, $\mathcal{L}_{\mathbf{y}}(2\theta_1^3 - \theta_1^2 \theta_2 + 3) = 2y_{3,0} - y_{2,1} + 3$. If $\mathbf{y}$ has a representing measure $\mu$, then $\mathcal{L}_{\mathbf{y}}$ simply maps polynomials $f$ to integrals of $f$ against $\mu$.

The key idea of the GMP approach is to convexify the problem by treating $\mathbf{y}$ as free variables and then introduce constraints to guarantee that $\mathbf{y}$ has a representing measure. First, let $\mathbf{v}_r(\boldsymbol{\theta}) := [\boldsymbol{\theta}^{\boldsymbol{\alpha}} : |\boldsymbol{\alpha}| \leq r] \in \mathbb{R}[\boldsymbol{\theta}]^{s(r)}$ be the vector of all $s(r)$ monomials of degree no greater than $r$. Then, define the *truncated moment matrix* as $\mathbf{M}_r(\mathbf{y}) := \mathcal{L}_{\mathbf{y}}(\mathbf{v}_r(\boldsymbol{\theta}) \mathbf{v}_r(\boldsymbol{\theta})^\top)$, where the linear functional $\mathcal{L}_{\mathbf{y}}$ is applied elementwise (see Example 3.1 below). If $\mathbf{y}$ has a representing measure $\mu$, then $\mathbf{M}_r(\mathbf{y})$ is simply a (positive) integral over rank 1 matrices $\mathbf{v}_r(\boldsymbol{\theta}) \mathbf{v}_r(\boldsymbol{\theta})^\top$ with respect to $\mu$, so necessarily

$\mathbf{M}_r(\mathbf{y}) \succeq 0$ holds. Furthermore, by Theorem 1 [25], for $\mathbf{y}$ to have a $K$-atomic representing measure, it is sufficient that $\text{rank}(\mathbf{M}_r(\mathbf{y})) = \text{rank}(\mathbf{M}_{r-1}(\mathbf{y})) = K$. So Problem 5 is equivalent to

$$
\begin{aligned}
\text{find} \quad & \mathbf{y} \in \mathbb{R}^{\mathbb{N}} \quad \text{(or equivalently, find } \mathbf{M}(\mathbf{y})) \\
\text{s.t.} \quad & \textstyle\sum_{\boldsymbol{\alpha}} a_{n\boldsymbol{\alpha}} y_{\boldsymbol{\alpha}} = \mathbb{E}[\phi_n(\mathbf{x})], \;\; n = 1, \ldots, N \\
& \mathbf{M}_r(\mathbf{y}) \succeq 0, \; y_{\mathbf{0}} = 1 \\
& \text{rank}(\mathbf{M}_r(\mathbf{y})) = K \text{ and } \text{rank}(\mathbf{M}_{r-1}(\mathbf{y})) = K.
\end{aligned}
\tag{6}
$$

Unfortunately, the rank constraints in Problem 6 are not tractable. We use the following relaxation to obtain our final (convex) optimization problem

$$
\begin{aligned}
\underset{\mathbf{y}}{\text{minimize}} \quad & \text{tr}(\mathbf{C}\mathbf{M}_r(\mathbf{y})) \\
\text{s.t.} \quad & \textstyle\sum_{\boldsymbol{\alpha}} a_{n\boldsymbol{\alpha}} y_{\boldsymbol{\alpha}} = \mathbb{E}[\phi_n(\mathbf{x})], \;\; n = 1, \ldots, N \\
& \mathbf{M}_r(\mathbf{y}) \succeq 0, \; y_{\mathbf{0}} = 1
\end{aligned}
\tag{7}
$$

where $\mathbf{C} \succ 0$ is a chosen scaling matrix. A common choice is $\mathbf{C} = \mathbf{I}_{s(r)}$ corresponding to minimizing the nuclear norm of the moment matrix, the usual convex relaxation for rank. Section A discusses some other choices of $\mathbf{C}$.

**Example 3.1** (moment matrix for a 1-dimensional Gaussian mixture). *Recall that the parameters $\boldsymbol{\theta} = [\xi, \sigma^2]$ are the mean and variance of a one dimensional Gaussian. Let us choose the monomials $\mathbf{v}_2(\boldsymbol{\theta}) = [1, \xi, \xi^2, \sigma^2]$. Step 4 for Figure 1 shows the moment matrix when using $r = 2$. Each row and column of the moment matrix is labeled with a monomial and entry $(i, j)$ is subscripted by the product of the monomials in row $i$ and column $j$. For $\phi_2(x) := x^2$, we have $f_2(\boldsymbol{\theta}) = \xi^2 + c$, which leads to the linear constraint $y_{2,0} + y_{0,1} - \mathbb{E}[x^2] = 0$. For $\phi_3(x) = x^3$, $f_3(\boldsymbol{\theta}) = \xi^3 + 3\xi c$, leading to the constraint $y_{3,0} + 3y_{1,1} - \mathbb{E}[x^3] = 0$.*

**Related work.** Readers familiar with the sum of squares and polynomial optimization literature [26, 27, 28, 29] will note that Problem 7 is similar to the SDP relaxation of a polynomial optimization problem. However, in typical polynomial optimization, we are only interested in solutions $\boldsymbol{\theta}^*$ that actually satisfy the given constraints, whereas here we are interested in $K$ solutions $[\boldsymbol{\theta}_k^*]_{k=1}^K$, whose *mixture* satisfies constraints corresponding to the moment conditions (3). Within machine learning, generalized PCA has been formulated as a moment problem [30] and the Hankel matrix (basically the moment matrix) has been used to learn weighted automata [13]. While similar tools are used, the conceptual approach and the problems considered are different. For example, the moment matrix of this paper consists of unknown moments of the model parameters, whereas exisiting works considered moments of the data that are always directly observable.

**Constraints.** Constraints such as non-negativity (for parameters which represent probabilities or variances) and parameter tying [31] are quite common in graphical models and are not easily addressed with existing method of moments approaches. The GMP framework allows us to incorporate some constraints using localizing matrices [32]. Thus, we can handle constraints during the estimation procedure rather than projecting back onto the constraint set as a post-processing step. This is necessary for models that only become identifiable by the observed moments after constraints are taken into account. We describe this method and its learning implications in Section C.1.

**Guarantees and statistical efficiency.** In some circumstances, e.g. in three-view mixture models or the mixture of linear regressions, the constraints fully determine the moment matrix – we consider these cases in Section 5 and Appendix B. While there are no general guarantee on Problem 7, the flat extension theorem tells us when the moment matrix corresponds to a unique solution (more discussions in Appendix A):

**Theorem 1** (Flat extension theorem [25]). *Let $\mathbf{y}$ be the solution to Problem 7 for a particular $r$. If $\mathbf{M}_r(\mathbf{y}) \succeq 0$ and $\text{rank}(\mathbf{M}_{r-1}(\mathbf{y})) = \text{rank}(\mathbf{M}_r(\mathbf{y}))$ then $\mathbf{y}$ is the optimal solution to Problem 6 for $K = \text{rank}(\mathbf{M}_r(\mathbf{y}))$ and there exists a unique $K$-atomic supporting measure $\mu$ of $\mathbf{M}_r(\mathbf{y})$.*

Recovering $\mathbf{M}_r(\mathbf{y})$ is linearly dependent on small perturbations of the input [33], suggesting that the method has polynomial sample complexity for most models where the moments concentrate at a polynomially rate. Finally, in Appendix C, we discuss a few other important considerations like noise robustness, making Problem 7 more statistical efficient, along with some technical results on the moment completion problem and some open problems.

# 4 Solution extraction

Having completed the (parameter) moment matrix $\mathbf{M}_r(\mathbf{y})$ (Section 3), we now turn to the problem of extracting the model parameters $[\boldsymbol{\theta}_k^*]_{k=1}^K$. The solution extraction method we present is based on ideas from solving multivariate polynomial systems where the solutions are eigenvalues of certain multiplication matrices [16, 17, 34, 35].[1] The main advantage of the solution extraction view is that higher-order moments and structure in parameters are handled in the framework without model-specific effort.

Recall that the true moment matrix is $\mathbf{M}_r(\mathbf{y}^*) = \sum_{k=1}^K \pi_k \mathbf{v}(\boldsymbol{\theta}_k^*)\mathbf{v}(\boldsymbol{\theta}^*{}_k)^\top$, where $\mathbf{v}(\boldsymbol{\theta}) := [\boldsymbol{\theta}^{\boldsymbol{\alpha}_1}, \ldots, \boldsymbol{\theta}^{\boldsymbol{\alpha}_{s(r)}}] \in \mathbb{R}[\boldsymbol{\theta}]^{s(r)}$ contains all the monomials up to degree $r$. We use $\boldsymbol{\theta} = [\theta_1, \ldots, \theta_P]$ for variables and $[\boldsymbol{\theta}_k^*]_{k=1}^K$ for the true solutions to these variables (note the boldface). For example, $\theta_{k,p}^* := (\boldsymbol{\theta}^*{}_k)_p$ denotes the $p^{\text{th}}$ value of the $k^{\text{th}}$ component, which corresponds to a solution for the variable $\theta_p$. Typically, $s(r) \gg K, P$ and the elements of $\mathbf{v}(\boldsymbol{\theta})$ are arranged in a degree ordering so that $||\boldsymbol{\alpha}_i||_1 \leq ||\boldsymbol{\alpha}_j||_1$ for $i \leq j$. We can also write $\mathbf{M}_r(y^*)$ as $\mathbf{M}_r(y^*) = \mathbf{VPV}^\top$, where the *canonical basis* $\mathbf{V} := [\mathbf{v}(\boldsymbol{\theta}_1^*), \ldots, \mathbf{v}(\boldsymbol{\theta}_K^*)] \in \mathbb{R}^{s(r) \times K}$ and $\mathbf{P} := \text{diag}(\pi_1, \ldots, \pi_K)$. At the high level, we want to factorize $M_r(y^*)$ to get $\mathbf{V}$, however we cannot simply eigen-decompose $\mathbf{M}_r(y^*)$ since $\mathbf{V}$ is not orthogonal. To overcome this challenge, we will exploit the internal structure of $\mathbf{V}$ to construct several other matrices that share the same factors and perform simultaneous diagonalization.

Specifically, let $\mathbf{V}[\boldsymbol{\beta}_1; \ldots; \boldsymbol{\beta}_K] \in \mathbb{R}^{K \times K}$ be a sub-matrix of $\mathbf{V}$ with only the rows corresponding to monomials with exponents $\boldsymbol{\beta}_1, \ldots, \boldsymbol{\beta}_K \in \mathbb{N}^P$. Typically, $\boldsymbol{\beta}_1, \ldots, \boldsymbol{\beta}_K$ are just the first $K$ monomials in $\mathbf{v}$. Now consider the exponent $\boldsymbol{\gamma}_p \in \mathbb{N}^P$ which is 1 in position $p$ and 0 elsewhere, corresponding to the monomial $\boldsymbol{\theta}^{\boldsymbol{\gamma}_p} = \theta_p$. The key property of the canonical basis is that multiplying each column $k$ by a monomial $\theta_{k,p}^*$ just performs a "shift" to another set of rows:

$$\mathbf{V}[\boldsymbol{\beta}_1; \ldots; \boldsymbol{\beta}_K]\,\mathbf{D}_p = \mathbf{V}\big[\boldsymbol{\beta}_1 + \boldsymbol{\gamma}_p; \ldots; \boldsymbol{\beta}_K + \boldsymbol{\gamma}_p\big], \quad \text{where } \mathbf{D}_p := \text{diag}(\theta_{1,p}^*, \ldots, \theta_{K,p}^*). \quad (8)$$

Note that $\mathbf{D}_p$ contains the $p^{\text{th}}$ parameter for all $K$ mixture components.

**Example 4.1** (Shifting the canonical basis). *Let $\boldsymbol{\theta} = [\theta_1, \theta_2]$ and the true solutions be $\boldsymbol{\theta}_1^* = [2, 3]$ and $\boldsymbol{\theta}_2^* = [-2, 5]$. To extract the solution for $\theta_1$ (which are $(\theta_{1,1}^*, \theta_{2,1}^*)$), let $\boldsymbol{\beta}_1 = (1, 0), \boldsymbol{\beta}_2 = (1, 1)$, and $\boldsymbol{\gamma}_1 = (1, 0)$.*

$$\mathbf{V} = \begin{array}{c} \\ 1 \\ \theta_1 \\ \theta_2 \\ \theta_1^2 \\ \theta_1\theta_2 \\ \theta_2^2 \\ \theta_1^2\theta_2 \end{array} \begin{array}{c} \mathbf{v}(\boldsymbol{\theta}_1) \quad \mathbf{v}(\boldsymbol{\theta}_2) \\ \begin{bmatrix} 1 & 1 \\ 2 & -2 \\ 3 & 5 \\ 4 & 4 \\ 6 & -10 \\ 9 & 25 \\ 12 & 20 \end{bmatrix} \end{array} \qquad \underbrace{\begin{array}{c} \theta_1 \\ \theta_1\theta_2 \end{array}\begin{array}{c}\mathbf{v}_1 \ \ \mathbf{v}_2\\\begin{bmatrix} 2 & -2 \\ 6 & -10 \end{bmatrix}\end{array}}_{\mathbf{V}[\boldsymbol{\beta}_1;\boldsymbol{\beta}_2]} \underbrace{\begin{bmatrix} 2 & 0 \\ 0 & -2 \end{bmatrix}}_{\text{diag}(\theta_{1,1},\theta_{2,1})} = \underbrace{\begin{array}{c} \theta_1^2 \\ \theta_1^2\theta_2 \end{array}\begin{array}{c}\mathbf{v}_1 \ \ \mathbf{v}_2\\\begin{bmatrix} 4 & 4 \\ 12 & 20 \end{bmatrix}\end{array}}_{\mathbf{V}[\boldsymbol{\beta}_1+\boldsymbol{\gamma}_1;\boldsymbol{\beta}_2+\boldsymbol{\gamma}_1]} \quad (9)$$

While the above reveals the structure of $\mathbf{V}$, we don't know $\mathbf{V}$. However, we recover its column space $\mathbf{U} \in \mathbb{R}^{s(r) \times K}$ from the moment matrix $\mathbf{M}_r(y^*)$, for example with an SVD. Thus, we can relate $\mathbf{U}$ and $\mathbf{V}$ by a linear transformation: $\mathbf{V} = \mathbf{UQ}$, where $\mathbf{Q} \in \mathbb{R}^{K \times K}$ is some unknown invertible matrix.

Equation 8 can now be rewritten as:

$$\mathbf{U}[\boldsymbol{\beta}_1; \ldots; \boldsymbol{\beta}_K]\mathbf{Q}\,\mathbf{D}_p = \mathbf{U}\big[\boldsymbol{\beta}_1 + \boldsymbol{\gamma}_p; \ldots; \boldsymbol{\beta}_K + \boldsymbol{\gamma}_p\big]\mathbf{Q}, \quad p = 1, \ldots, P, \quad (10)$$

which is a generalized eigenvalue problem where $\mathbf{D}_p$ are the eigenvalues and $\mathbf{Q}$ are the eigenvectors. Crucially, the eigenvalues, $\mathbf{D}_p = \text{diag}(\theta_{1,p}^*, \ldots, \theta_{K,p}^*)$ give us solutions to our parameters. Note that for any choice of $\boldsymbol{\beta}_1, \ldots, \boldsymbol{\beta}_K$ and $p \in [P]$, we have generalized eigenvalue problems that share eigenvectors $Q$, though their eigenvectors $D_p$ may differ. Corresponding eigenvalues (and hence solutions) can be obtained by solving a simultaneous generalized eigenvalue problem, e.g., by using random projections like Algorithm B of [4] or more robust [37] simultaneous diagonalization algorithms [38, 39, 40].

Table 2: Applications of the Polymom framework. See Appendix B.2 for more details.

**Mixture of linear regressions**

| **Model** | **Observation functions** |
|---|---|
| $\mathbf{x} = [x, \upsilon]$ is observed where $x \in \mathbb{R}^D$ is drawn from an unspecified distribution and $\upsilon \sim \mathcal{N}(\mathbf{w} \cdot x, \sigma^2 I)$, and $\sigma$ is known. The parameters are $\boldsymbol{\theta}_k^* = (\mathbf{w}_k) \in \mathbb{R}^D$. | $\phi_{\boldsymbol{\alpha},b}(\mathbf{x}) = x^{\boldsymbol{\alpha}} \upsilon^b$ for $0 \le |\boldsymbol{\alpha}| \le 3, b \in [2]$. **Moment polynomials** $f_{\boldsymbol{\alpha},1}(\boldsymbol{\theta}) = \sum_{p=1}^{P} \mathbb{E}[x^{\boldsymbol{\alpha}+\boldsymbol{\gamma}_p}] w_p$ $f_{\boldsymbol{\alpha},2}(\boldsymbol{\theta}) = \mathbb{E}[x^{\boldsymbol{\alpha}}]\sigma^2 + \sum_{p,q=1}^{P} \mathbb{E}[x^{\boldsymbol{\alpha}} x_p x_q] w_p w_q$, where the $\boldsymbol{\gamma}_p \in \mathbb{N}^P$ is 1 in position $p$ and 0 elsewhere. |

**Mixture of Gaussians**

| **Model** | **Observation functions** |
|---|---|
| $\mathbf{x} \in \mathbb{R}^D$ is observed where $\mathbf{x}$ is drawn from a Gaussian with diagonal covariance: $\mathbf{x} \sim \mathcal{N}(\boldsymbol{\xi}, \mathrm{diag}(\mathbf{c}))$. The parameters are $\boldsymbol{\theta}_k^* = (\boldsymbol{\xi}_k, \mathbf{c}_k) \in \mathbb{R}^{D+D}$. | $\phi_{\boldsymbol{\alpha}}(\mathbf{x}) = \mathbf{x}^{\boldsymbol{\alpha}}$ for $0 \le |\boldsymbol{\alpha}| \le 4$. **Moment polynomials** $f_{\boldsymbol{\alpha}}(\boldsymbol{\theta}) = \prod_{d=1}^{D} h_{\alpha_d}(\xi_d, c_d).$ [2] |

**Multiview mixtures**

| **Model** | **Observation functions** |
|---|---|
| With 3 views, $\mathbf{x} = [x^{(1)}, x^{(2)}, x^{(3)}]$ is observed where $x^{(1)}, x^{(2)}, x^{(3)} \in \mathbb{R}^D$ and $x^{(\ell)}$ is drawn from an unspecified distribution with mean $\boldsymbol{\xi}^{(\ell)}$ for $\ell \in [3]$. The parameters are $\boldsymbol{\theta}_k^* = (\boldsymbol{\xi}_k^{(1)}, \boldsymbol{\xi}_k^{(2)}, \boldsymbol{\xi}_k^{(3)}) \in \mathbb{R}^{D+D+D}$. | $\phi_{ijk}(\mathbf{x}) = x_i^{(1)} x_j^{(2)} x_k^{(3)}$ where $1 \le i, j, k \le D$. **Moment polynomials** $f_{ijk}(\boldsymbol{\theta}) = \xi_i^{(1)} \xi_j^{(2)} \xi_k^{(3)}.$ |

We describe one approach to solve (10), which is similar to Algorithm B of [4]. The idea is to take $P$ random weighted combinations of the equations (10) and solve the resulting (generalized) eigende-composition problems. Let $R \in \mathbb{R}^{P \times P}$ be a random matrix whose entries are drawn from $\mathcal{N}(0,1)$. Then for each $q = 1, \dots Q$, solve $\mathbf{U}[\boldsymbol{\beta}_1; \dots; \boldsymbol{\beta}_K]^{-1} \left( \sum_{p=1}^{P} R_{q,p} \mathbf{U}[\boldsymbol{\beta}_1 + \boldsymbol{\gamma}_p; \dots; \boldsymbol{\beta}_K + \boldsymbol{\gamma}_p] \right) \mathbf{Q} = \mathbf{Q}\mathbf{D}_q$. The resulting eigenvalues can be collected in $\Lambda \in \mathbb{R}^{P \times K}$, where $\Lambda_{q,k} = \mathbf{D}_{q,k,k}$. Note that by definition $\Lambda_{q,k} = \sum_{p=1}^{P} R_{q,p} \boldsymbol{\theta}_{k,p}^*$, so we can simply invert to obtain $[\boldsymbol{\theta}_1^*, \dots, \boldsymbol{\theta}_K^*] = R^{-1}\Lambda$. Although this simple approach does not have great numerical properties, these eigenvalue problems are solvable if the eigenvalues $[\lambda_{q,1}, \dots, \lambda_{q,K}]$ are distinct for all $q$, which happens with probability 1 as long as the parameters $\boldsymbol{\theta}_k^*$ are different from each other.

In Appendix B.1, we show how a prior tensor decomposition algorithm from [4] can be seen as solving Equation 10 for a particular instantiation of $\boldsymbol{\beta}_1, \dots \boldsymbol{\beta}_K$.

## 5 Applications

Let us now look at some applications of Polymom. Table 2 presents several models with corresponding observation functions and moment polynomials. It is fairly straightforward to write down observation functions for a given model. The moment polynomials can then be derived by computing expectations under the model– this step can be compared to deriving gradients for EM.

We implemented Polymom for several mixture models in Python (code: https://github.com/sidaw/polymom). We used CVXOPT to handle the SDP and the random projections algorithm from to extract solutions. In Table 3, we show the relative error $\max_k ||\boldsymbol{\theta}_k - \boldsymbol{\theta}_k^*||_2/||\boldsymbol{\theta}_k^*||_2$ averaged over 10 random models of each class.

In the rest of this section, we will discuss guarantees on parameter recovery for each of these models.

| | Methd. | EM | TF | Poly | EM | TF | Poly | EM | TF | Poly |
|---|---|---|---|---|---|---|---|---|---|---|
| **Gaussians** | $K, D$ | | $T = 10^3$ | | | $T = 10^4$ | | | $T = 10^5$ | |
| spherical | 2, 2 | **0.37** | 2.05 | 0.58 | **0.24** | 0.73 | 0.29 | 0.19 | 0.36 | **0.14** |
| diagonal | 2, 2 | **0.44** | ~~2.15~~ | 0.48 | 0.48 | ~~4.03~~ | **0.40** | 0.38 | ~~2.46~~ | **0.35** |
| constrained | 2, 2 | 0.49 | ~~7.52~~ | **0.38** | 0.47 | ~~2.56~~ | **0.30** | 0.34 | ~~3.02~~ | **0.29** |
| **Others** | $K, D$ | | $T = 10^4$ | | | $T = 10^5$ | | | $T = 10^6$ | |
| 3-view | 3, 3 | **0.38** | 0.51 | 0.57 | 0.31 | 0.33 | **0.26** | 0.36 | 0.16 | **0.12** |
| lin. reg. | 2, 2 | - | - | **3.51** | - | - | **2.60** | - | - | **2.52** |

Table 3: $T$ is the number of samples, and the error metric is defined above. **Methods:** EM: sklearn initialized with k-means using 5 random restarts; TF: tensor power method implemented in Python; Poly: Polymom by solving Problem 7. **Models:** for mixture of Gaussians, we have $\sigma \approx 2||\mu_1 - \mu_2||_2$. spherical and diagonal describes the type of covariance matrix. The mean parameters of constrained Gaussians satisfies $\mu_1 + \mu_2 = 1$. The best result is **bolded**. TF only handles spherical variance, but it was of interest to see what TF does if the data is drawn from mixture of Gaussians with diagonal covariance, these results are in ~~strikeout~~.

**Mixture of Linear Regressions.** We can guarantee that Polymom can recover parameters for this model when $K \leq D$ by showing that Problem 6 can be solved exactly: observe that while no entry of the moment matrix $\mathbf{M}_3(\mathbf{y})$ is directly observed, each observation gives us a linear constraint on the entries of the moment matrix and when $K \leq D$, there are enough equations that this system admits an unique solution for $\mathbf{y}$.

Chaganty et al. [9] were also able to recover parameters for this model under the same conditions ($K \leq D$) by solving a series of low-rank tensor recovery problems, which ultimately requires the computation of the same moments described above. In contrast, the Polymom framework makes the dependence on moments upfront and takes care of the heavy-lifting in a problem-agnostic manner. Lastly, the model can be extended to handle per component noise by including $\sigma$ as a parameter, an extension that is not possible using the method in [9].

**Multiview Mixtures.** We can guarantee parameter recovery when $K \leq D$ by proving that Problem 7 can be solved exactly (see Section B.2).

**Mixture of Gaussians.** In this case however, the moment conditions are non-trivial and we cannot guarantee recovery of the true parameters. However, Polymom is guaranteed to recover a mixture of Gaussians that match the moments. We can also apply constraints to the model: consider the case of 2d mixture where the mean parameters for all components lies on a parabola $\xi_1 - \xi_2^2 = 0$. In this case, we just need to add constraints to Problem 7: $y_{(1,0)+\boldsymbol{\beta}} - y_{(0,2)+\boldsymbol{\beta}} = 0$ for all $\boldsymbol{\beta} \in \mathbb{N}^2$ up to degree $|\boldsymbol{\beta}| \leq 2r - 2$. By incorporating these constraints at estimation time, we can possibly identify the model parameters with less moments. See Section C for more details.

## 6 Conclusion

We presented an unifying framework for learning many types of mixture models via the method of moments. For example, for the mixture of Gaussians, we can apply the same algorithm to both mixtures in 1D needing higher-order moments [3, 11] and mixtures in high dimensions where lower-order moments suffice [6]. The Generalized Moment Problem [15, 14] and its semidefinite relaxation hierarchies is what gives us the generality, although we rely heavily on the ability of nuclear norm minimization to recover the underlying rank. As a result, while we always obtain parameters satisfying the moment conditions, there are no formal guarantees on consistent estimation. The second main tool is solution extraction, which characterizes a more general structure of mixture models compared the tensor structure observed by [6, 4]. This view draws connections to the literature on solving polynomial systems, where many techniques might be useful [35, 18, 19]. Finally, through the connections we've drawn, it is our hope that Polymom can make the method of moments as turnkey as EM on more latent-variable models, as well as improve the statistical efficiency of method of moments procedures.

**Acknowledgments.** This work was supported by a Microsoft Faculty Research Fellowship to the third author and a NSERC PGS-D fellowship for the first author.

## Footnotes

[1] [36] is a short overview and [35] is a comprehensive treatment including numerical issues.

[2] $h_\alpha(\xi, c) = \sum_{i=0}^{\lfloor \alpha/2 \rfloor} a_{\alpha, \alpha-2i} \xi^{\alpha-2i} c^i$ and $a_{\alpha,i}$ be the absolute value of the coefficient of the degree $i$ term of the $\alpha^{\text{th}}$ (univariate) Hermite polynomial. For example, the first few are $h_1(\xi, c) = \xi$, $h_2(\xi, c) = \xi^2 + c$, $h_3(\xi, c) = \xi^3 + 3\xi c$, $h_4(\xi, c) = \xi^4 + 6\xi^2 c + 3c^2$.

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
