[Supplementary Material]


[31] D. Koller and N. Friedman. *Probabilistic graphical models: principles and techniques*. MIT Press, 2009.

[32] R. Curto and L. Fialkow. The truncated complex K-moment problem. *Transactions of the American mathematical society*, 352(6):2825–2855, 2000.

[33] R. W. Freund and F. Jarre. A sensitivity result for semidefinite programs. *Operations Research Letters*, 32:126–132, 2004.

[34] R. M. Corless, P. M. Gianni, B. M. Trager, and S. M. Watt. The singular value decomposition for polynomial systems. In *International Symposium on Symbolic and Algebraic Computation*, pages 195–207, 1995.

[35] H. J. Stetter. *Numerical polynomial algebra*. Siam, 2004.

[36] P. Dreesen, K. Batselier, and B. D. Moor. Back to the roots: Polynomial system solving, linear algebra, systems theory. In *IFAC Symposium on System Identification (SYSID)*, pages 1203–1208, 2012.

[37] V. Kuleshov, A. Chaganty, and P. Liang. Simultaneous diagonalization: the asymmetric, low-rank, and noisy settings. *arXiv*, 2015.

[38] J. Cardoso and A. Souloumiac. Jacobi angles for simultaneous diagonalization. *SIAM Journal on Matrix Analysis and Applications*, 17(1):161–164, 1996.

[39] A. Bunse-Gerstner, R. Byers, and V. Mehrmann. Numerical methods for simultaneous diagonalization. *SIAM Journal on Matrix Analysis and Applications*, 14(4):927–949, 1993.

[40] B. Afsari. Simple LU and QR based non-orthogonal matrix joint diagonalization. In *Independent Component Analysis and Blind Signal Separation*, pages 1–7, 2006.

# A Theory of the Moment Completion Problem

For solution extraction, we assumed that moments of all monomials are observed but for many models only polynomials of parameters can be estimated from the data. For example, in a Gaussian mixture the $2^{nd}$ moment observable function $\phi(x) = \xi^2 + c$ is a polynomial, but solution extraction requires moments of monomials like $\xi^2$ and $c$. Furthermore, we assumed in Section 4 that there exists underlying true parameters $[\boldsymbol{\theta}_k^*]_{k=1}^K$ while an arbitrary moment sequence of the parameters $\mathbf{y}$ and its corresponding moment matrix $\mathbf{M}(\mathbf{y})$ may not correspond to any parameters (i.e. no representing measure). In Section 5, we give an example of how moment completion can be done with just linear algebra for multiview models, and we focus on the harder case of having to solve the SDP Problem 7 in this section.

While we do not have a complete answer since the rank constrained Problem 6 cannot be solved, we analyze the relaxed moment completion problem (Problem 7) and give some sufficient conditions for solution extraction and sufficient conditions for parameter recovery.

## A.1 Conditions for solution extraction

Under the exact setting, and assuming a moment matrix generated by true parameters $\boldsymbol{\theta}^*$, In Section 4, we showed that simple conditions based only on the column space basis is sufficient for solution extraction to be successful. However, to further investigate consistency and noise, we need to address a few more important issues. First consider the noiseless setting, we may not have enough moment contraints to guarantee a unique solution (identifiability). Even if we assume that we have enough constraints for identifying a $K$ mixture, we still do not know if solving the relaxed Problem 7 that relaxed the $\mathrm{rank} = K$ constraint can recover the true parameters. Second, under noise, there may not exist a rank $K$ basis of the moment matrix and even when a rank $K$ basis exists, it may not correspond to any true parameters.

We now address the problem of when some moment matching parameters can be extracted. We first discuss the concept of a *flat extension* which is the same as conditions in Section 4 where "$\mathbf{B}_p := \mathbf{U}\big[\boldsymbol{\gamma}_p + [\boldsymbol{\beta}_1, \ldots, \boldsymbol{\beta}_K]\big]$ is observed" and $\mathbf{U}[\boldsymbol{\beta}_1; \ldots; \boldsymbol{\beta}_K]$ is a column space basis of $\mathbf{M}_r(\mathbf{y})$. Let the highest degree monomial of $\mathbf{U}[\boldsymbol{\beta}_1; \ldots; \boldsymbol{\beta}_K]$ be of degree $r - 1 = \deg(\boldsymbol{\theta}^{\boldsymbol{\beta}_K}) = |\boldsymbol{\beta}_K|$, and the highest degree monomial of $\mathbf{B}_p := \mathbf{U}\big[\boldsymbol{\gamma}_p + [\boldsymbol{\beta}_1, \ldots, \boldsymbol{\beta}_K]\big]$ be of degree $r = |\boldsymbol{\gamma}_p + \boldsymbol{\beta}_K| = |\deg(\boldsymbol{\beta}_K)| + 1$. Since $\mathbf{U}[\boldsymbol{\beta}_1; \ldots; \boldsymbol{\beta}_K]$ is a basis of $\mathrm{col}(\mathbf{M}_r(\mathbf{y}))$

$$\mathrm{rank}\left(\mathbf{M}_{r-1}(\mathbf{y})\right) = \mathrm{rank}\left(\mathbf{U}[\boldsymbol{\beta}_1; \ldots; \boldsymbol{\beta}_K]\right) = K \tag{11}$$

$$= \mathrm{rank}\left(\mathbf{M}_r(\mathbf{y})\right) \geq \mathrm{rank}\left(\mathbf{U}\big[\boldsymbol{\gamma}_p + [\boldsymbol{\beta}_1, \ldots, \boldsymbol{\beta}_K]\big]\right). \tag{12}$$

If we got this basis from the moment matrix, then we say that the moment matrix $\mathbf{M}_{r-1}(\mathbf{y})$ corresponding to $\mathbf{U}[\boldsymbol{\beta}_1; \ldots; \boldsymbol{\beta}_K]$ has a flat extension, because $\mathbf{M}_{r-1}(\mathbf{y})$ can be extended to a moment matrix $\mathbf{M}_r(\mathbf{y})$ with higher degree monomials without an increase in rank. The concept of flat extension and its consequences are of central importance for the truncated moment problem, which is quite relevant to our problem and studied extensively by [1, 2, 3, 4]. Next, we reproduce the simplest flat extension theorem:

**Theorem A.1** ([1]: flat extension theorem)**.** *Suppose* $\mathbf{M}_{r-1}(\mathbf{y}) \succeq 0$ *and there exists* $\mathbf{M}_r(\mathbf{y})$ *so that* $\mathrm{rank}(\mathbf{M}_r(\mathbf{y})) = \mathrm{rank}(\mathbf{M}_{r-1}(\mathbf{y}))$ *(i.e. a flat extension), then there exists an unique* $\mathrm{rank}(\mathbf{M}_r(\mathbf{y}))$*-atomic representing measure* $\mu$ *of* $\mathbf{M}_r(\mathbf{y})$.

Here the first column of $\mathbf{M}_r(\mathbf{y})$ contains every monomial of degree up to $r$ so that $\deg(\mathbf{v}_r(\boldsymbol{\theta})) = r$. However, several generalizations of the flat extension theorem are also useful for estimation of mixture models where sparse monomials are handled [5, 6] or where constraints are handled [4].

The conceptual importance is that Theorem A.1 allows us to work with just the moment matrix satisfying constraints from possibly noisy observations, without assuming the moment matrix is generated by some true parameters. Of course, it also provides a checkable criterion for when solutions can be extracted [7]. We still do not know if solving Problem 7 provides a flat extension in a finite number of steps. [8, 9, 10] investigated this issue very recently and showed that linear optimization over the cone of moments have finite convergence under generic conditions (theorem 4.2 of [10]).

Still, our issue is not fully resolved as representing measures under linear constraints may not be unique, and as a result even a flat moment matrix may not correspond to the true parameters. For parameter fitting, we'd like to find the solution with minimal rank or otherwise optimal in some way. We explore this issue next but unfortunately we can only give some partial answers.

**Proposition A.2** (existence of $\mathbf{C}$)**.** *In the noiseless setting, there exist* $\mathbf{C}$ *so that minimizing* $\mathbf{C} \bullet \mathbf{M}_r(\mathbf{y})) = \mathbf{c} \cdot \mathbf{y}$ *will give the right solution.*

*Proof.* Let $\mathbf{M}_r(\mathbf{y}) = \mathbf{U}\boldsymbol{\Sigma}\mathbf{U}^\mathsf{T}$ be the SVD with $\mathbf{U} \in \mathbb{R}^{s(r) \times K}$ and $\boldsymbol{\Sigma} \in \mathbb{R}^{K \times K}$. Let $\mathbf{U}_\perp \in \mathbb{R}^{s(r) \times (s(r) - K)}$ be the orthogonal compliment of $\mathbf{U}$, then any $\mathbf{C} = \mathbf{U}_\perp \mathbf{D} \mathbf{U}_\perp^\mathsf{T}$ is fine for arbitrary diagonal matrix $\mathbf{D} \in \mathbb{R}^{(s(r)-K) \times (s(r)-K)}$.

The convex iteration algorithm [11] is one way to reduce rank that sometimes works for us empirically, where if the convex iteration algorithm converges to 0, then the moment matrix has rank $K$.

# B Examples

## B.1 Tensor factorization as solution extraction

**Example B.1** (Tensor decomposition as solution extraction). *Many latent variable models have been tackled via tensor decomposition [12], and symmetric, undercomplete tensor decomposition can be framed as a solution extraction problem. Suppose we observe the tensor $\mathbf{T} := \sum_{k=1}^{K} \boldsymbol{\theta}^{*}{}_{k}^{\otimes 3} \in \mathbb{R}^{P \times P \times P}$. We would like to recover the components $\boldsymbol{\theta}_{k}^{*}$. For us, the inputs are constraints $\theta_r \theta_s \theta_t - T_{rst} = 0$ for all $r, s, t = 1, \ldots, P$. Choose $\mathbf{v}(\boldsymbol{\theta}) = [1, \theta_1, \ldots, \theta_P, \theta_1^2, \theta_1 \theta_2, \ldots, \theta_P^2] = [1, \boldsymbol{\theta}, \mathrm{vecs}(\boldsymbol{\theta} \otimes \boldsymbol{\theta})]$, where $\mathrm{vecs} : \mathbb{R}^{P \times P} \to \mathbb{R}^{P^2}$ just flattens the matrix. In the simplest case, suppose $P = K$ and $\mathrm{rank}(\mathbf{U}) = K$. Then the fully observed $\mathbf{U}$ is*

$$
\mathbf{U} = \begin{matrix} \scriptstyle{size} \\ \scriptstyle 1 \\ \scriptstyle P \\ \scriptstyle P^2 \end{matrix} \overset{P}{\begin{bmatrix} \mathbf{U}_1 \\ \mathbf{U}_2 \\ \mathbf{U}_3 \end{bmatrix}} = \begin{matrix} \scriptstyle{terms} \\ \scriptstyle 1 \\ \scriptstyle \boldsymbol{\theta} \\ \scriptstyle \mathrm{vecs}(\boldsymbol{\theta} \otimes \boldsymbol{\theta}) \end{matrix} \overset{\boldsymbol{\theta}}{\begin{bmatrix} \mathscr{L}_{\mathbf{y}}(\boldsymbol{\theta}) \\ \mathscr{L}_{\mathbf{y}}(\boldsymbol{\theta} \otimes \boldsymbol{\theta}) \\ \mathscr{L}_{\mathbf{y}}(\mathrm{vecs}(\boldsymbol{\theta} \otimes \boldsymbol{\theta}) \otimes \boldsymbol{\theta}) \end{bmatrix}} \tag{13}
$$

*where the linear functional $\mathscr{L}_{\mathbf{y}}$ applies elementwise. One choice of basis is just all the variables $\mathbf{U}[\boldsymbol{\beta}_1; \ldots; \boldsymbol{\beta}_K] = \mathbf{U}_2$ and the eigenvalue problem we are required to solve is the generalized Hermitian eigenvalue problem $\mathbf{U}_2 \mathbf{Q} \mathbf{D} = \left( \sum_{p=1}^{P} \eta_p \mathscr{L}_{\mathbf{y}}(\theta_p \boldsymbol{\theta} \otimes \boldsymbol{\theta}) \right) \mathbf{Q}$. [13] proposed an algorithm that is procedurally identical, where, in their notation $\mathrm{Pairs} := \mathbf{U}_2$ and $\mathrm{Triples}(\eta) := \left( \sum_{p=1}^{P} \eta_p \mathscr{L}_{\mathbf{y}}(\theta_p \boldsymbol{\theta} \otimes \boldsymbol{\theta}) \right)$, and the algorithm proposed needed to solve the eigenvalue problem $B(\eta) = \mathrm{Pairs}^{-1} \mathrm{Triples}(\eta)$.*

Typically, $\boldsymbol{\beta}_1, \ldots, \boldsymbol{\beta}_K$ are just the first $K$ monomials in $\mathbf{v}$ (i.e. the $K$ monomials of the smallest degree).

Under this formulation, generalization to the fully-observed overcomplete tensor decomposition case $K \geq D = P$ is clear if we observe enough moments to have enough basis vectors such that $\mathrm{rank}(\mathbf{U}[\boldsymbol{\beta}_1; \ldots; \boldsymbol{\beta}_K]) = K$:

**Proposition B.2.** *If $K \leq 1 + P + P^2 + \cdots + P^r = \frac{P^{r+1}-1}{P-1}$, then solution extraction succeeds if we observe moments up to order $2r + 1$ and monomials vectors of the true parameters $\mathbf{v}_r(\boldsymbol{\theta}_1), \ldots, \mathbf{v}_r(\boldsymbol{\theta}_K)$ are linearly independent.*

*Proof.* To get the theoretical result, it suffices to consider higher-order moments:

$$
\mathbf{U} = \begin{matrix} \scriptstyle{terms} \\ \scriptstyle \mathrm{vecs}(\boldsymbol{\theta}^{\otimes r}) \\ \scriptstyle \mathrm{vecs}(\boldsymbol{\theta}^{\otimes r+1}) \end{matrix} \overset{\mathrm{vecs}(\boldsymbol{\theta}^{\otimes r})}{\begin{bmatrix} \mathscr{L}_{\mathbf{y}}\big(\mathrm{vecs}(\boldsymbol{\theta}^{\otimes r}) \otimes \mathrm{vecs}(\boldsymbol{\theta}^{\otimes r})\big) \\ \mathscr{L}_{\mathbf{y}}\big(\mathrm{vecs}(\boldsymbol{\theta}^{\otimes r+1}) \otimes \mathrm{vecs}(\boldsymbol{\theta}^{\otimes r})\big) \end{bmatrix}} \tag{14}
$$

where we can take the $\mathbf{U}[\boldsymbol{\beta}_1; \ldots; \boldsymbol{\beta}_K]$ from the top block, and $\mathbf{U}\big[\boldsymbol{\beta}_1 + \boldsymbol{\gamma}_q; \ldots; \boldsymbol{\beta}_K + \boldsymbol{\gamma}_q\big]$ belongs to the bottom block for all $q$. So $2r + 1$ order moments is needed if $K \leq P^r$ and this result is comparable to [14]. In practice, we would take all moments $\mathrm{vecs}(\boldsymbol{\theta}^{\otimes 1}), \ldots, \mathrm{vecs}(\boldsymbol{\theta}^{\otimes r+1})$. We may use lower order moments as well:

$$
\mathbf{U} = \begin{matrix} \scriptstyle{terms} \\ \scriptstyle \mathrm{vecs}(\boldsymbol{\theta}^{\otimes 1}) \\ \scriptstyle \mathrm{vecs}(\boldsymbol{\theta}^{\otimes 2}) \\ \scriptstyle \vdots \\ \scriptstyle \mathrm{vecs}(\boldsymbol{\theta}^{\otimes r+1}) \end{matrix} \overset{\mathrm{vecs}(\boldsymbol{\theta}^{\otimes 1}) \quad \mathrm{vecs}(\boldsymbol{\theta}^{\otimes 2}) \quad \cdots \quad \mathrm{vecs}(\boldsymbol{\theta}^{\otimes r})}{\begin{bmatrix} & \vdots & \\ \cdots & \mathscr{L}_{\mathbf{y}}(\mathrm{vecs}(\boldsymbol{\theta}^{\otimes l}) \otimes \mathrm{vecs}(\boldsymbol{\theta}^{\otimes m})) & \cdots \\ & \vdots & \end{bmatrix}} \tag{15}
$$

where the entry of this matrix at block $l, m$ is $\mathscr{L}_{\mathbf{y}}(\mathrm{vecs}(\boldsymbol{\theta}^{\otimes l}) \otimes \mathrm{vecs}(\boldsymbol{\theta}^{\otimes m}))$ as expected. While this still requires observing $2r + 1^{th}$ order moments, lower order moments are more accurate and can result in better parameter estimates. $\square$

## B.2 Moment completion for specific models

### B.2.1 Mixture of Linear Regressions

In Example 2.2, we described the mixture of linear regressions model in 1-dimension with parameters $\boldsymbol{\theta}_k^* = (w_k, \sigma_k^2)$. Let us now consider the $D$-dimensional extension: we observe $\mathbf{x} = [x, v]$[3] where $x := [x_1, \ldots, x_D]$

is drawn from an unspecified distribution and $\upsilon = w \cdot x + \epsilon$ with $\epsilon \sim \mathcal{N}(0, \sigma^2)$ for a known $\sigma$. The parameters are $\boldsymbol{\theta}_k^* = (w_k)$ for $1 \leq k \leq K$. Next, we choose observation functions $\phi_{\boldsymbol{\alpha}, b}(\mathbf{x}) = x^{\boldsymbol{\alpha}} \upsilon^b$ for $\boldsymbol{\alpha} : 0 \leq |\boldsymbol{\alpha}| \leq 3$ and $0 \leq b \leq 3$, with corresponding moment polynomials: $f_{\boldsymbol{\alpha}, b}(\boldsymbol{\theta}, \mathbf{x}) = x^{\boldsymbol{\alpha}} \mathbb{E}_{\epsilon \sim \mathcal{N}(0, \sigma^2)} \left[ (w \cdot x + \epsilon)^b \right]$. These polynomials can be expressed in closed form using Hermite polynomials (see Section B.2.2). For example, $f_{\mathbf{0}, 2}(\boldsymbol{\theta}, \mathbf{x}) = \left( (w \cdot x)^2 + \sigma^2 \right)$.

Given these observation functions and moment polynomials, and data, the Polymom framework solves the moment completion problem (Problem 6) followed by solution extraction (Section 4) to recover the parameters. Further, we can guarantee that Polymom can recover parameters for this model when $K \leq D$ by showing that Problem 6 can be solved exactly. Note that while no entry of the moment matrix is directly observed, each observation gives us a linear constraint on the entries of the moment matrix. Let $\boldsymbol{\gamma}_p \in \mathbb{N}^P$ be the vector with value 1 at position $p$ and 0 elsewhere, then $\mathscr{L}_{\mathbf{y}}(f_{\boldsymbol{\alpha}, 1}(\boldsymbol{\theta})) = \sum_{p=1}^{P} \mathbb{E}[x^{\boldsymbol{\alpha} + \boldsymbol{\gamma}_p}] y_{\boldsymbol{\gamma}_p}$, and $\mathscr{L}_{\mathbf{y}}(f_{\boldsymbol{\alpha}, 2}(\boldsymbol{\theta})) = \left( \mathbb{E}[x^{\boldsymbol{\alpha}}] \sigma^2 + \sum_{p,q=1}^{P} \mathbb{E}[x^{\boldsymbol{\alpha} + \boldsymbol{\gamma}_p + \boldsymbol{\gamma}_q}] y_{\boldsymbol{\gamma}_p + \boldsymbol{\gamma}_q} \right)$, etc. When $K \leq D$, there are enough equations that this system admits an unique solution for $\mathbf{y}$.

Note that [15] recover parameters for this model by solving a series of low-rank tensor recovery problems, which ultimately requires the computation of the same moments described above. In contrast, the Polymom framework makes the dependence on moments upfront and takes care of the heavy-lifting in a problem-agnostic manner. Furthermore, we can even obtain parameters outside the regime of [15]: with the above observation functions and moment polynomials, we can recover parameters (with a certificate) .

### B.2.2 Mixture of Gaussians

We now look at $D$-dimensional extensions to Example 2.1. Let the data be drawn from Gaussians with diagonal covariance, $x|z \sim \mathcal{N}(\xi_z, \text{diag}(c_z))$. The parameters of this model are $\boldsymbol{\theta}_k^* = (\xi_k, c_k) \in \mathbb{R}^{2D}$. The observable functions are $\phi_{\boldsymbol{\alpha}}(\mathbf{x}) := \mathbf{x}^{\boldsymbol{\alpha}}$, and the moment polynomials are $f_{\boldsymbol{\alpha}}(\boldsymbol{\theta}) = \mathbb{E}[\mathbf{x}^{\boldsymbol{\alpha}}] = \prod_{d=1}^{D} h_{\alpha[d]}(\xi[d], c[d])$, where $h_{\alpha}(\xi, c) = \sum_{i=0}^{\lfloor \alpha/2 \rfloor} a_{\alpha, \alpha - 2i} \xi^{\alpha - 2i} c^i$ and $a_{\alpha, i}$ be the absolute value of the coefficient of the degree $i$ term of the $\alpha^{\text{th}}$ (univariate) Hermite polynomial. The first few are $h_1(\xi, c) = \xi$, $h_2(\xi, c) = \xi^2 + c$, $h_3(\xi, c) = \xi^3 + 3\xi c$, $h_4(\xi, c) = \xi^4 + 6\xi^2 c + 3c^2$.

Using this set of $\phi_{\boldsymbol{\alpha}}$ and $f_{\boldsymbol{\alpha}}$, Polymom will attempt to solve the SDP in Problem 7 and recover the parameters. In this case however, the moment conditions are non-trivial and we cannot guarantee recovery of the true parameters. However, Polymom is guaranteed to recover parameters that match the moments and that minimizes nuclear norm.

We can modify this model by introducing constraints: consider the case of 2D mixture where the mean parameters for all components lies on a parabola $\xi_1 - \xi_2^2 = 0$. In this case, we just need to add constraints to Problem 7: $y_{(1,0) + \boldsymbol{\beta}} - y_{(0,2) + \boldsymbol{\beta}} = 0$ for all $\boldsymbol{\beta} \in \mathbb{N}^2$ up to degree $|\boldsymbol{\beta}| \leq 2r - 2$.

By incorporating these contraints at estimation time, we can possibly identify the model parameters with less moments. See Section C for more details.

### B.2.3 Multiview Mixtures

Here we consider the three-view mixture model which has been well studied in [12, section 3.3]. We will show that we can solve the model without explicit whitening, a transformation that has been shown to introduce noise[16]. The model is a mixture of three conditionally independent arbitrary distributions parameterized by their conditional means: we have $z \sim \text{Multinomial} \, \pi$, $\mathbf{x}_l | z \sim p_l(\boldsymbol{\xi}_z^{(l)})$ where $p_l(\boldsymbol{\xi}_z^{(l)})$ is such that $E_{\mathbf{x}_l | z}[\mathbf{x}_l] = \boldsymbol{\xi}$. The parameters are $\boldsymbol{\theta}_k = [\boldsymbol{\xi}^{(1)}, \boldsymbol{\xi}^{(2)}, \boldsymbol{\xi}^{(3)}]$. Using the observation functions $\phi = [x^{(1)}, x^{(2)}, x^{(3)}, x^{(1)} \otimes x^{(2)}, \ldots, x^{(1)} \otimes x^{(2)} \otimes x^{(3)}]$, we have the following moment polynomials, $f = [\boldsymbol{\xi}^{(1)}, \boldsymbol{\xi}^{(2)}, \boldsymbol{\xi}^{(3)}, \boldsymbol{\xi}^{(1)} \otimes \boldsymbol{\xi}^{(2)}, \ldots, \boldsymbol{\xi}^{(1)} \otimes \boldsymbol{\xi}^{(2)} \otimes \boldsymbol{\xi}3]$.

The multiview mixture model is another model for which we can guarantee parameter recovery when $K \leq D$. To prove this is the case, we will again show that Problem 7 can be solved exactly. It suffices to consider just the first $P$ columns of the moment matrix $\mathbf{M}_2$, which are almost directly observable. As before, $\text{vecs}(\cdot)$ just flattens a matrix into a vector.

$$
\mathbf{M}_2^{\mathsf{T}} = 
\begin{array}{c}
\\ \boldsymbol{\xi}_1 \\ \boldsymbol{\xi}_2 \\ \boldsymbol{\xi}_3
\end{array}
\begin{bmatrix}
\boldsymbol{\xi}_1 & \boldsymbol{\xi}_2 & \boldsymbol{\xi}_3 & \text{vecs}(\boldsymbol{\xi}_1 \otimes \boldsymbol{\xi}_2) & \text{vecs}(\boldsymbol{\xi}_1 \otimes \boldsymbol{\xi}_3) & \text{vecs}(\boldsymbol{\xi}_2 \otimes \boldsymbol{\xi}_3) \\
Z_{2,0,0} & \mathbf{Y}_{1,1,0} & \mathbf{Y}_{1,0,1} & Z_{2,1,0} & Z_{2,0,1} & \mathbf{Y}_{1,1,1} \\
\mathbf{Y}_{1,1,0} & Z_{0,2,0} & \mathbf{Y}_{0,1,1} & Z_{1,2,0} & \mathbf{Y}_{1,1,1} & Z_{0,2,1} \\
\mathbf{Y}_{1,0,1} & \mathbf{Y}_{0,1,1} & Z_{0,0,2} & \mathbf{Y}_{1,1,1} & Z_{1,0,2} & Z_{0,1,2}
\end{bmatrix}
\tag{16}
$$

where $Y_{\alpha_1, \alpha_3, \alpha_3}$ and $Z_{\alpha_1, \alpha_3, \alpha_3}$ are both equal to $\mathscr{L}_{\mathbf{y}}(\boldsymbol{\xi}_1^{\otimes \alpha_1} \otimes \boldsymbol{\xi}_2^{\otimes \alpha_2} \otimes \boldsymbol{\xi}_3^{\otimes \alpha_3})$, but are used to respectively denote observed and unknown variables. However, this equation is only partially true as both sides contain the

same set of values but the precise arrangements depends on where the minor matrix appears in the moment matrix. We ignore this problem as it should be clear from the row and column labels. In the undercomplete case, it is assumed that $\text{rank}(\mathbf{U}) = K \leq \min(P_1, P_2, P3)$, thus we can easily complete this matrix using simple linear algebra in the exact case by repeatedly applying Lemma B.3 below. Generally, we may try to complete the moment matrix by solving Problem 7 from these partial observations, provided that optimizing with the nuclear norm recovers the true rank.

**Lemma B.3** (low rank completion of missing corner). *For any matrix $\Gamma = \begin{bmatrix} A & B \\ C & X \end{bmatrix}$ with a missing block $X$, where $\text{rank}(\Gamma) = \text{rank}(A) = \text{rank}(B) = K$ and $A \in \mathbb{R}^{K \times K}$, $X = CA^{-1}B$ uniquely completes $\Gamma$.*

*Proof.* Because $A$ contains the entire $K$ elements basis, there exists unique $Y, Z \in \mathbb{R}^{K \times K}$ so that $B = AY$ and $C = ZA$. Similarly, $X = ZB = CA^{-1}B$. $\qed$

# C  Extensions

## C.1  Constraints on parameters

Constraints on parameters is a common and important consideration in applications. While constraints can often be addressed in maximum likelihood or *maximum a prioterior* learning using EM [17, see shared parameters], it is less clear how to address constraints under the tensor decomposition approach because of its reliance on special tensor structure and it is well-known that MME generally can give us parameters outside of the parameter space even in the well-specified case.

**Example C.1.** *Examples of constraints on parameters **Some parameters are known**: Gaussian with sparse covariance matrix where we already know that some dimensions are uncorrelated; to solve a substitution cipher using an HMM, the transitions matrix is a language model that is given.*

***Parameters are tied**: transitions in an HMM might only depend on the relatively difference between states if the states are ordered i.e. the transition matrix is Toeplitz.*

***Polytope constraints**: some of our parameters might be probabilities: $\boldsymbol{\theta} = [\pi_1, \ldots, \pi_P, \xi_1, \ldots]$, $\pi_p \geq 0$, $\sum_{p=1}^{P} \pi_p = 1$ e.g. prior parameter in LDA.*

***Semialgebraic constraints**: For some polynomial $g \in \mathbb{R}[\boldsymbol{\theta}]$, $g_i(\boldsymbol{\theta}_k^*) \geq 0, i = 1, \ldots, I$. Quite powerful, can express things like discrete sets $\boldsymbol{\theta}_i \in \{0, 1\}$, elipsoids.*

The obvious attempt is to project to the feasible set after computing an unconstrained estimation with MME. But this approach has several serious issues. First, some constrained models are only identifiable after the constraints are taken into account, which happens when the model has a lot of parameters and we cannot observe correspondingly more moments. In this case, unconstrained estimation is useful only if we can characterize the entire subset of the parameters space satisfying moment conditions, which is generally not possible in the tensor decomposition approach. Second, we need to determine what projection to use. In the case of two equal parameters, if one estimate is much more noisy than the other, it can be better to just ignore the more noisy estimate than to project under the wrong metric (see Example C.3). Third and strangely, even in the case when the first two issues are handled, it was observed by [18] for probablities parameters, that clipping to 0 is empirically inferior compared to heuristics like taking the absolute value, which is not a projection.

Under the Polymom formulation, we can take constraints into account during estimation. The technique of localizing matrix [3] in moment theory allows us to deal with semialgebraic constraints. Of course, the computational complexity increases if the constraints are themselves complicated and high degree. Next, we define the localization matrix, give an example, and then give a constrained version of the flat extension theorem.

**Example C.2** (localizing matrix for an inequality constraint). *Let $\boldsymbol{\theta} = [c, \xi]$, so that $\boldsymbol{\theta}^{\boldsymbol{\alpha}} = c^{\alpha_1}\xi^{\alpha_2}$ and $\mathscr{L}_{\mathbf{y}}(\boldsymbol{\theta}^{\boldsymbol{\alpha}}) = y_{\boldsymbol{\alpha}}$, and chose the monomials $\mathbf{v}_2(\boldsymbol{\theta}) = [1, c, \xi, c^2, c\xi, \xi^2]$. Suppose that $c$ is the variance and we want to have constraint that $c - 1 \geq 0$, then*

$$
\mathbf{M}_1((c-1)\mathbf{y}) = 
\begin{matrix}
 & \overset{1}{} & \overset{c}{} & \overset{\xi}{} \\
\begin{matrix} 1 \\ c \\ \xi \end{matrix} &
\left[ \begin{matrix}
y_{1,0} - 1 & y_{2,0} - y_{1,0} & y_{1,1} - y_{1,0} \\
y_{2,0} - y_{1,0} & y_{3,0} - y_{2,0} & y_{2,1} - y_{2,0} \\
y_{1,1} - y_{0,1} & y_{2,1} - y_{1,1} & y_{1,2} - y_{0,2}
\end{matrix} \right]
\end{matrix}
\tag{17}
$$

*it is clear that a necessary condition for extracted solutions to satisfy the constraint $c - 1 \geq 0$ is that $\mathbf{M}_1((c-1)\mathbf{y}) \succeq 0$ since $\boldsymbol{f}^\mathsf{T}\mathbf{M}_1((c-1)\mathbf{y})\boldsymbol{f} = \mathscr{L}_{\mathbf{y}}(f(\boldsymbol{\theta})^2(c-1)) \geq 0$.*

## C.2 Noise and statistical efficiency

In the presense of noise Problem 7 may not be feasible and even if it was, it may not be ideal to exactly match noisy moments. Furthermore, it is argued that higher order moments are too noisy to be useful, but there are also more of them and they do contain more information about the model parameters as long as we can model how noisy they are. We consider the problem with slack $\epsilon$ and a weighting matrix $\mathbf{W} \succ 0 \in \mathbb{R}^{N \times N}$ modelling how much noise is present in each constraint function. This effect is fairly well-known, and here is a very simple example which shows that even much more noisy measurements can improve efficiency.

**Example C.3** (efficient estimation). *Suppose* $X \sim \mathcal{N}([\xi, \xi], \mathrm{diag}[\sigma^2, c\sigma^2])$ *and we would like to estimate the mean parameter $\xi$ by matching moments. Any estimators of the form $\hat{\xi} = \frac{1}{T}\sum_{t=1}^{T}(\gamma x_{t,1} + (1-\gamma)x_{t,2})$ are consistent and has risk*

$$R = \mathbb{E}\left[(\hat{\xi} - \xi)^2\right] = \mathbb{E}\left[\left(\gamma\sum_{t=1}^{T}\mathbf{x}_{t,2} - \gamma\xi + (1-\gamma)\sum_{t=1}^{T}\mathbf{x}_{t,2} - (1-\gamma)\xi\right)^2\right] \tag{18}$$

$$= \mathbb{E}\left[\gamma^2\left(\xi - \frac{1}{T}\sum_{t=1}^{T}\mathbf{x}_{t,1}\right)^2 + (1-\gamma)^2\left(\xi - \frac{1}{T}\sum_{t=1}^{T}\mathbf{x}_{t,2}\right)^2\right] \tag{19}$$

$$= \frac{1}{T}(\gamma^2\sigma^2 + (1-\gamma)^2 c\sigma^2) \tag{20}$$

*under the squared loss, and the efficient estimator would have $\gamma = \frac{c-1}{c}$ and a risk of $\frac{\sigma^2}{T}\frac{c^2-c+1}{c^2}$. For $c = 10$, the risk for efficient estimation is $0.91\frac{\sigma^2}{T}$ whereas for $\gamma = 0.5$, the risk is $2.75\frac{\sigma^2}{T}$.*

$$\begin{array}{ll}
\underset{\mathbf{g},\mathbf{y}}{\text{minimize}} & \mathbf{C} \bullet \mathbf{M}(\mathbf{y}) \\
\text{s.t.} & g_n = \sum_{\boldsymbol{\alpha}} a_{n\boldsymbol{\alpha}} y_{\boldsymbol{\alpha}} - \mathbb{E}[\phi_n(\mathbf{x})] \\
& \boldsymbol{g}^{\mathsf{T}}\mathbf{W}\boldsymbol{g} \leq \epsilon \\
& \mathbf{M}(\mathbf{y}) \succeq 0
\end{array} \tag{21}$$

In the simplest case when $\mathbf{W} = \mathbf{I}_N$, and $\epsilon = 0$, Problem 21 is the same as Problem 7.

$$\begin{array}{ll}
\underset{\mathbf{g},\mathbf{y}}{\text{minimize}} & \mathbf{C} \bullet \mathbf{M}(\mathbf{y}) \\
\text{s.t.} & g_n = \sum_{\boldsymbol{\alpha}} a_{n\boldsymbol{\alpha}} y_{\boldsymbol{\alpha}} - \mathbb{E}[\phi_n(\mathbf{x})] \\
& \mathbf{W} \bullet \mathbf{F} \leq \epsilon \\
& \mathbf{M}(\mathbf{y}) \succeq 0 \\
& \begin{bmatrix} 1 & g^{\mathsf{T}} \\ g & \mathbf{F} \end{bmatrix} \succeq 0
\end{array} \tag{22}$$

A good weighting matrix $\mathbf{W}$ should put more weights on moment conditions that can be estimated more precisely. The asymptotically efficient weighting matrix suggested by the *Generalized Method of Moments* [19] is

$$\mathbf{W}^{-1} = \mathbb{E}\left[\boldsymbol{g}([\boldsymbol{\theta}_k]_{k=1}^{K}, \mathbf{x})\boldsymbol{g}([\boldsymbol{\theta}_k]_{k=1}^{K}, \mathbf{x})^{\mathsf{T}}\right] \approx \frac{1}{T}\sum_{t=1}^{T}\boldsymbol{g}([\boldsymbol{\theta}_k]_{k=1}^{K}, \mathbf{x})\boldsymbol{g}([\boldsymbol{\theta}_k]_{k=1}^{K}, \mathbf{x})^{\mathsf{T}} \tag{23}$$

**Theorem 2** (Gen.MM is asymptotically efficient [19]). *Let $g_n(\boldsymbol{\theta}, \boldsymbol{X}) := \sum_k f_n(\boldsymbol{\theta}_k) - h_n(\boldsymbol{X})$ so that $\mathbb{E}[h_n(\boldsymbol{X})] = \mathbb{E}[\phi_n(\mathbf{x})]$. Let $W^{-1} = \mathbb{E}[\mathbf{g}(\boldsymbol{\theta}, \boldsymbol{X})\mathbf{g}(\boldsymbol{\theta}, \boldsymbol{X})^{\mathsf{T}}] \approx \frac{1}{T}\sum_{t=1}^{T}\mathbf{g}(\boldsymbol{\theta}, \boldsymbol{X}_t)\mathbf{g}(\boldsymbol{\theta}, \boldsymbol{X}_t)^{\mathsf{T}}$ Iterative Gen.MM is efficient with this weighting matrix $\mathbf{W}$.*

# D Separability

For conditional models, the coefficients of the moment polynomials can depend on the data but such dependence can sometimes break the process of converting from component moment constraints to mixture moment constraints. In this section, we define separability, which is a sufficient condition on what dependence is allowed under Polymom and then we give some counterexamples.

Consider a mixture of linear regressions [20, 15], where the parameters $\boldsymbol{\theta}_k = (w_k, \sigma_k^2)$ are the slope and noise variance for each component $k$. Then each data point $\mathbf{x} = [x, y]$ is drawn from component $k$ by sampling $x$ from an unknown distribution *independent* of $k$ and setting $y = w_k x + \epsilon$, where $\epsilon \sim \mathcal{N}(0, \sigma_k^2)$. If we take

observation function $\phi_{b,c}(\mathbf{x}) = x^b y^c$, then the corresponding $f_{b,c}(\boldsymbol{\theta})$ depends on the unknown distribution of $x$: for example, $f_{1,2}(\boldsymbol{\theta}) = \mathbb{E}[x^3]w + \mathbb{E}[x]\sigma^2$. In contrast, for the mixture of Gaussians, we had $f_2(\boldsymbol{\theta}) = \mu^2 + \sigma^2$, which only depends on the parameters.

However, not all is lost, since the key thing is that $f_{1,2}(\boldsymbol{\theta})$ depends only on the distribution of $x$, which is independent of the component $k$ and furthermore can be estimated from data. More generally, we allow $f_n$ to depend on $\mathbf{x}$ but in a restricted way. We say that $f_n(\boldsymbol{\theta}, \mathbf{x})$ is *separable* if $\mathbb{E}[f_n(\boldsymbol{\theta}, \mathbf{x})]$ does not depend on the parameters $[\boldsymbol{\theta}_k]_{k=1}^K$ of the mixture generating $\mathbf{x}$. In other words,

$$\mathbb{E}[\phi_n(\mathbf{x})] = \mathbb{E}[f_n(\boldsymbol{\theta}, \mathbf{x})] \text{ where for all } k: \ \mathbb{E}_{\mathbf{x}|z=k}[f(\boldsymbol{\theta}; \mathbf{x})] = f(\boldsymbol{\theta}). \ \mathbb{E}[f_n(\boldsymbol{\theta}, \mathbf{x}) \mid z = k] = \mathbb{E}[f_n(\boldsymbol{\theta}, \mathbf{x})] \in \mathbb{R}[\boldsymbol{\theta}].$$
(24)

In this case, we can define $f_n(\boldsymbol{\theta}) := \mathbb{E}[f_n(\boldsymbol{\theta}, \mathbf{x})]$, and (3) is still valid. For the mixture of linear regressions, we would define $f_{b,c}(\boldsymbol{\theta}, \mathbf{x}) = x^b \, \mathbb{E}_{\epsilon \sim \mathcal{N}(0,\sigma^2)}[(wx + \epsilon)^c]$. In this more general setup, the approximate moment equations on $T$ data points is $\frac{1}{T} \sum_{t=1}^T [f_n(\boldsymbol{\theta}, \mathbf{x}_t)] = \frac{1}{T} \sum_{t=1}^T \phi_n(\mathbf{x}_t)$.

An example of non-separability is a mixture of linear regressions where the variance is not a parameter and is different across mixture components: $\boldsymbol{\theta} = (w)$ and $\mathbf{x} = (x, y)$. Recall that $\mathbb{E}[xy^2] = \sum_{k=1}^K \pi_k (\mathbb{E}[x^3]w_k^2 + \mathbb{E}[x]\sigma_k^2)$, but $\mathbb{E}[x^3]w_k^2 + \mathbb{E}[x]\sigma_k^2$ cannot be written as $\mathbb{E}[f_n(w_k, \mathbf{x})]$ for any $f_n$, since it depends on $\sigma_k^2$. Thus, this example falls outside our framework. In the simplest case, we can make $f_n(w, \mathbf{x})$ separable by introducing $\sigma_k$ as a parameter, but this is not always possible if the noise distribution is unknown or if $\sigma_k(x)$ depends on $x$. For example, if we have heteroskedastic noise, $\mathbb{E}[x^a(y - w \cdot x)] = 0$ are valid moment constraints for individual components, but it is not clear how to convert this to the mixture case.

## Footnotes

[3] We use $v$ here since $y$ is reserved for the parameter moments.