[Reviews · NeurIPS 2015]

Submitted by Assigned_Reviewer_1

This paper is about mixture of polynomials, a new technique of method of moments in parameter estimation in mixture models. It provides a general recipe: first define N observation functions {\phi_n}_{n = 1}^N; then compute its conditional expectation with respect to each mixture component in terms of udnerlying parameters; then solve optimization problem (6), or its convex relaxation (7). Experiments shows the improved recovery accuracy.

Quality: This approach is technically sound up to Equation (6) -- it is not clear that the convex relaxation provide accurate parameter recovery, (it is pointed out in Appendix that it is even not clear if it provides a flat extension.) Also, it will be good if the algorithm were provided with sample complexity and recovery guarantees.

For the parameter recovery step, it seems like we can find several tuples of (\beta_1, ...,\beta_k) and perform simultaneous diagonalization. This may make the algorithm more robust.

Clarity: I find the non-separability example in page 3 quite confusing. Why cannot the algorithm recover both w_k and \sigma_k simultaneously?

Example 4.2 does not directly use optimization problem (6) or (7), but rather discusses the approach used in paper [6], which I think is unnecessary. Could the authors explain?

Originality: this paper is interesting, since it brings new techniques into the general method of moments; this is quite different from existing work in tensor methods and numerical linear algebra.

EDIT: the idea of using SDP for tensor decomposition has also been explored by [1]. It would be good if the authors can provide comparisons to that work.

Significance: overall, the paper brings interesting new insights of the traditional method of moments, specifically, for the problem of parameter estimation in mixture models.

Reference: [1] Tang and Shah, Guaranteed Tensor Decomposition: A Moment Approach, ICML 2015.
Summary: This paper is about mixture of polynomials, a new technique of method of moments in parameter estimation of mixture models. It brings interesting new insights into the traditional method of moments.

Submitted by Assigned_Reviewer_2

The paper presents a general framework for deriving methods of moments for learning mixture models. Its main contributions are: (1) showing how expressions for the moments of the base distribution can be bootstrapped to derive expressions for the moments of mixtures of base distributions; and, (2) providing recipes for solving the resulting moment equations combining SDP and generalised eigenvalue problems. The overall impression is that the paper does not contain a great deal of technical novelty, but it provides a fresh perspective on moment matching methods for mixture models. The main paper is well-written, but several crucial details are discussed only in the appendix.

It is a bit disappointing that the paper pays very little attention to two aspects that initially attracted a lot of attention about spectral methods of moments: the possibility of proving finite-sample bounds, and their scalability as compared to vanilla EM. About the latter, my impression is that obtaining really efficient solutions for the SDP problems arising from different mixture models will require exploiting specific structural properties in each individual case. So the method might not be as "turnstyle" as the authors promise when scalability considerations are taken into account.

The scope of the experiments is quite limited, but nearly enough due to the nature of the paper. It does not help dispel any doubts about the scalability of the proposed method, but it showcases the potential benefits of this approach when compared to EM. There are, however, two aspects of these results that could be improved: adding a comparison with the baseline from reference [9] for the linear regression case, and comparing the methods using more than one accuracy measure (eg. perplexity), since somehow one suspects that L2 distance to the true parameter is what poly is hiddenly optimizing, while EM is clearly optimizing another metric.

The weakest point of the paper is perhaps its lack of awareness of some relevant previous results. In particular, although the paper does a good job at connecting with literature related to generalised moment problems and solutions of polynomial equations, it misses some quite direct connections to other methods of moments based on Hankel matrices. These follow by simply observing that the moment matrix M(y) is nothing else than a Hankel matrix with indices on the commutative free monoid \mathbb{N}^P. From this point of view, rank properties of M(y), the way in which the truncation M_r(y) is defined, and the shifting exploited in Section 4, are all direct consequences of the same properties for Hankel matrices on the non-commutative free monoid (eg. see [1]). Also, non-commutative versions of problems (6) and (7) were considered already in [2] and [3] with similar constraints as the ones used in the paper, and in [4] in a more general non-probabilistic setting. In view of this, the paper would probably do a better service to the community by making these connections explicit.

After reading the authors' response I am more inclined to recognize the novelty in their contribution. However, the feeling of this reviewer (and apparently other reviewers as well) is that in general the paper glosses over a number of important points (eg. identifiability, sample bounds, scalability of SDP relaxations, possibility of special-purpose optimization algorithms). It is clear that a conference paper cannot be asked to tackle all these in detail, but the current paper could be improved with a more systematic and upfront discussion of what it tries and tries not to do. In summary, I think this work shows promise, but perhaps it could benefit from a more elaborate presentation.

*** Minor Comment ***

--- In Line 193, the only immediate conclusion one can draw from \mu being K-atomic is that rank(M_r(y)) <= K (note the inequality, not equality like in the paper). To obtain the equality claimed in the paper one needs to: (1) suppose that r is large enough, and (2) somehow make an argument showing that when the theta^*_k are all different then the vectors v_r(theta^*_k) for k = 1...K are linearly independent. Since this point might have impact on the identifiability of the model, I think it is too important to be hand-waved like the authors do.

[1] Borja Balle, Xavier Carreras, Franco M. Luque, Ariadna Quattoni: Spectral learning of weighted automata - A forward-backward perspective. Machine Learning 96(1-2): 33-63 (2014)

[2] Raphael Bailly, Xavier Carreras, Ariadna Quattoni: Unsupervised Spectral Learning of Finite State Transducers. NIPS 2013

[3] Raphael Bailly, Xavier Carreras, Franco M. Luque, Ariadna Quattoni: Unsupervised Spectral Learning of WCFG as Low-rank Matrix Completion. EMNLP 2013

[4] Borja Balle and Mehryar Mohri: Spectral learning of general weighted automata via constrained matrix completion. NIPS 2012

Summary: The overall impression is that the paper does not contain a great deal of technical novelty, but it provides a somewhat fresh perspective on moment matching methods for mixture models. There is room for improvement in establishing connections and comparing with prior related work.

Submitted by Assigned_Reviewer_3

The paper would benefit from considering sampling noise and identifiability issues. The computational complexity and sample complexity are not discussed. This analysis would improve the paper. The authors suggest learning in recursive manner which suffers from an obvious obstacle: error propagation. Hence, the aforementioned analysis would help the paper substantially.
Summary: The paper uses method of moments to learn different mixture models. Specifically, the paper looks as polynomial observation functions.

Submitted by Assigned_Reviewer_4

The paper frames the MoM estimation of mixture models firstly as finding a particular probability measure on the parameter space constrained to satisfy the moment conditions (5), then secondly as finding a rank-constrained semi-definite parameter moment matrix satisfying the moment conditions (6), which is relaxed when necessary to minimizing the nuclear norm of the matrix (7). Once the moment matrix is computed, the model parameters are extracted by solving a simultaneous generalized eigenvalue problem on shifts of an orthonormal basis of the column space.

The paper borrows from multiple disciplines (method of moments, optimization, computer algebra) to generalize the MoM to handle arbitrary moments and mixture models. Although there is no attempt to establish consistency or identifiability of the recovered parameters, the paper concretely extends the existing (e.g., Hsu et al.) MoM techniques for recovering latent variable models with a new, unified framework.

The paper is generally well-written, but it's also very dense (especially the main sections 3 and 4). Much of the materials critical for understanding the presented work is brushed aside as a pointer to other works or in the supplementary sections. There are some issues with the notation as well:

- Define \delta in the main paper. - Define f(\theta) as well as f_n(\theta).

- Stick to either \mu^*(\theta) or \mu^*(d \theta). - "w" -> "w^2" in f_{1,2}(\theta) = E[x^3] w + E[x] \sigma^2 (Section 2.1) In general, I think it'd help the reader a lot if the paper can supply a brief introduction to GMP.

My concern is that it seems difficult for the method to have significant practical values. With relaxation, it's arguably another heuristic like EM for estimating mixture models, but in that case I find EM to be a much more easy-to-use and empirically effective choice. Consistency - the usual strength of MoM techniques - not necessarily present, it feels like a method no more general than EM but suffering more severely from numerical instability. (Though the paper supplies positive experimental results, I have suspicions about their validity.)
Summary: The paper extends existing techniques on method-of-moment (MoM) estimation of latent variable models with a unified framework based on a generalized moment problem (GMP) followed by parameter extraction.

Submitted by Assigned_Reviewer_5

Conditions, say for GMMs, under which approximation guarantees on the parameters hold would be very interesting, though seemingly difficult. Novelty relative to "GPCA with Denoising: A Moments-Based Convex Approach" by Ozay et al should be addressed.
Summary: A method is proposed for learning mixtures of parametric distributions based on the method of moments by approximately solving the problem of finding a mixing measure and corresponding mixture component parameters which satisfy the moment equations via an SDP relaxation and a subsequent extraction step. Since the idea isn't completely novel, it would be good to see more development of the theory (approximation/estimation guarantees).

Submitted by Assigned_Reviewer_6

This paper poses the problem of estimating the parameters of a mixture model and proposes a solution based on the method of moments as they apply to polynomial optimization subject to polynomial constraints (Laurent,2008). More specifically, the problem is posed as that of recovering an atomic measure that satisfies certain given moment constraints.

This is relaxed to minimizing the trace of the corresponding moment matrix subject to its positive-definiteness (which is a necessary condition for a matrix to be a moment matrix). The latter is a semidefinite program. Then the authors show how to obtain the mixture parameters by means of a collection of generalized eigenvalue problems.

In my view the key contribution of the paper is problems (6) and (7), which show how to pass from the estimation of the mixture parameters to a semidefinite program. However, problem (7) is a special instance of a polynomial optimization problem subject to polynomial constraints. The methodology that the authors adopt to solve that problem is an agglomeration of what seem to be known techniques. So on one hand the connection with mixture estimation is nice and novel, on the other hand the methodology is clearly presented, yet not particularly novel. Consequently, there is no need for the paper to be so densely written, having a size much larger than that of a typical NIPS paper (the authors use same font for the main text as for the abstract).

I also think the authors should address the following question: What happens when the solution of the relaxed problem (7) is full rank (which is often the case in practice)? I suppose the authors keep the first K principal components of the moment matrix M and then proceed to do solution extraction as in section 4? Also, what if K is unknown?
Summary: The main contribution of the paper is that of passing from the problem of estimating mixture parameters to a semidefinite program together with a collection of generalized eigenvalue problems. The paper is interesting, yet it is too dense.

Author Feedback
Author rebuttal: Prior works using the Hankel matrices to learn weighted finite state automata: We thank reviewer 2 for providing a connection to these prior works and interpreting our work in that context. While we were somewhat aware of this line of work, we did not see a clear connection since both the language and the motivation seemed quite different on the surface. We agree that ignoring this connection is a missed opportunity and we will compare and contrast with this line of work in the revised version of our paper following the reviewer's suggestions. However, we do not think this nice connection diminishes our main contribution of extending method of moments parameter recovery to a wider range of mixture models since it seems non-trivial to extend these prior works similarly.

Identifiability: Identifiability is a delicate issue for the generalized moment problem (as discussed in [21]) and thus it is hard to specify transparent identifiability criteria for our method in the general case beyond flat extension. However, for specific model classes already discussed in the literature, such as those described in sections 5.1 (mixture of linear regressions) and 5.3 (multiview mixtures), it is shown that known regimes (when k <= d) trivially imply flat extension and thus identifiability in our framework. We will elaborate on this discussion and connections between flat extension and identifiability in a revised version of the paper.

Sample complexity: A detailed perturbative analysis is tedious, but it is easy to show that if the model is identifiable using moments of rank up to r, then the sample complexity of our method is polynomial. To see this, note that the two steps of our method, moment completion and solution extraction are both numerically stable-- the former is a semidefinite program (in the worst case), which has a linear dependence on small perturbations in the input (see [39]) and the latter simply involves solving a simultaneous diagonalization problem which has been shown to have linear dependence on perturbation in the input [4, 36]. Finally, the perturbation in the moment conditions ($E[\phi_n(x)]$) depends on the distribution of the data, but except in degenerate cases (e.g. Cauchy), is polynomially dependent on the number of samples via the central limit theorem.

[39] Roland W. Freund, Florian Jarre, A sensitivity result for semidefinite programs, Operations Research Letters, Volume 32, Issue 2, March 2004. (http://www.sciencedirect.com/science/article/pii/S0167637703000695)

Specific Reviewer Questions:
R1:
Q: Why cannot the algorithm recover both w_k and \sigma_k?
A: We can in fact recover \sigma_k if it is specified as an actual model parameter. However, we defined the set of parameter as \theta = (w) as was the case in [9]. The set up is described in the paragraph starting at line 128. We will work on making this paragraph more clear.
Q: why Example 4.2
A: This was included because many models have been solved using tensor factorization and we thought it would be good to draw a direct connection from tensor factorization to the solution extraction step while skipping the less studied SDP step. The example also shows how existing methods can be seen as a special case of our framework. We agree that this connection is interesting mostly to people with prior exposure to the tensor factorization line of work and example 4.2 might be better suited in the appendix.

R5:
A: We agree that sampling noise and identifiability are important: this paper focuses on a different problem and there is no space to address every important issue (we already have complaints that the current writing is too dense). Our experiments (though limited in number) show how well the method works in the presence of sampling noise and we have discussed some issues related to noise in the appendix. While it is very nice that some works on spectral learning can address identifiability, it should also be acceptable to consider learning separately (you can run EM on non-identifiable models): for non-identifiable models (i.e. when flat extension does not hold) our method will still give a moment matching set of parameters that depends on the choice of objective. If we get the opportunity, we will address these important considerations in an extended version. (Also see our general response on identifiability above).

R7
A: We will try to make this more clear. There is a recipe for this issue, which is discussed in [13] and [14] as well as the appendix. We increase r, and solve the next problem in the hierarchy until we get the flat extension at which point you can recover a moment matching solution of rank r. Conditions for obtaining flat extension has been studied fairly extensively. This is what we would do in the case of unknown K. However, the general problem of controlling the rank of SDP is unresolved.